# Progress in Antiviral Fullerene Research

**DOI:** 10.3390/nano12152547

**Published:** 2022-07-24

**Authors:** Piao-Yang Xu, Xiao-Qing Li, Wei-Guang Chen, Lin-Long Deng, Yuan-Zhi Tan, Qianyan Zhang, Su-Yuan Xie, Lan-Sun Zheng

**Affiliations:** 1College of Chemistry and Chemical Engineering, Xiamen University, Xiamen 361005, China; 20520170155073@stu.xmu.edu.cn (P.-Y.X.); yuanzhi_tan@xmu.edu.cn (Y.-Z.T.); syxie@xmu.edu.cn (S.-Y.X.); lszheng@xmu.edu.cn (L.-S.Z.); 2Funano New Material Technology Company Ltd., Xiamen 361110, China; fjlxq0708@126.com (X.-Q.L.); cweig123@163.com (W.-G.C.); 3Pen-Tung Sah Institute of Micro-Nano Science and Technology, Xiamen University, Xiamen 361005, China; denglinlong@xmu.edu.cn

**Keywords:** fullerene, water-soluble fullerene derivatives, antivirus, nanodrug

## Abstract

Unlike traditional small molecule drugs, fullerene is an all-carbon nanomolecule with a spherical cage structure. Fullerene exhibits high levels of antiviral activity, inhibiting virus replication in vitro and in vivo. In this review, we systematically summarize the latest research regarding the different types of fullerenes investigated in antiviral studies. We discuss the unique structural advantage of fullerenes, present diverse modification strategies based on the addition of various functional groups, assess the effect of structural differences on antiviral activity, and describe the possible antiviral mechanism. Finally, we discuss the prospective development of fullerenes as antiviral drugs.

## 1. Introduction

Fullerenes are all-carbon molecules discovered in 1985 [1]. They are spherical or ellipsoidal in shape, with a hollow cage structure. Three discoverers of fullerene C_60_ won the Nobel Prize in chemistry in 1996. Fullerene C_60_, the representative fullerene, is ~0.7 nm in diameter. In the past 30 years, with the continuous development of fullerene preparation technology [2,3,4,5], fullerenes have presented unprecedented opportunities in the fields of biomedicine, catalysis, superconduction, and photovoltaics. Nanomolecules have important applications in cancer treatment, diagnosis, imaging, drug delivery, catalysis, and biosensing [6,7,8,9,10,11,12,13,14,15]. Fullerene molecules not only have defined nanostructures, but also unique electronic characteristics, photophysical properties, and excellent biocompatibility. Fullerene molecules have properties that differ from those of traditional small molecule drugs, which make fullerenes nanodrug candidates [16], especially for diagnosis and treatment. For example, fullerenes and their derivatives can be used as antioxidants against inflammatory diseases, due to their rich conjugated double bonds, which scavenge free radicals [17,18]. Fullerene C_60_ activates tumor immunity by polarizing tumor-associated macrophages and combines with immune checkpoint inhibitors (PD-L1 monoclonal antibody) to achieve efficient tumor immunotherapy [19]. Fullerene C_70_ derivatives, as photosensitizers, produce singlet oxygen, which can effectively kill tumor cells [20]. Endohedral metal fullerenes serve as new nuclear magnetic resonance contrast agents [21,22] for treating liver steatosis [23] and tumors [24,25,26,27,28,29,30]. Additionally, some fullerene derivatives stabilize immune effector cells and prevent/inhibit the release of proinflammatory mediators. Therefore, they are potential drugs for a variety of diseases, such as asthma [31], arthritis [32], and multiple sclerosis [33]. Moreover, carboxylic acid derivatives of fullerenes can cut DNA under visible light irradiation, with the potential for use as photosensitive biochemical probes [34]. Fullerene C_60_ carboxylic acid derivatives also exhibit neuroprotective activity [35] and strongly inhibit tumor growth in a zebrafish xenograft model [36]. Because nanoparticles have been approved as drugs and drug carriers, fullerenes have great potential as drugs or gene delivery carriers [37,38].

Currently, more than 90% of infectious diseases in humans are caused by viruses. The most well-known include the influenza virus, human immunodeficiency virus (HIV), and Ebola virus, which have caused serious damage [39,40,41,42,43]. Although several anti-HIV and anti-Ebola drugs, such as saquinavir, ritonavir, T20, lopinavir, ribavirin, tenofovir, and remdesivir, have been developed, their efficacy is not satisfactory. Severe acute respiratory syndrome (SARS), which broke out in China in 2003, is a respiratory infection caused by a coronavirus. So far, there is no specific medicine for SARS. The novel coronavirus, SARS-CoV-2, now circulating worldwide, is more infectious than SARS or HIV. For patients infected with SARS-CoV-2, there are no specific antiviral drugs.

Fullerenes and their derivatives exert significant inhibitory effects against HIV [39], herpes simplex virus (HSV) [40], influenza virus [44], Ebola virus [45], cytomegalovirus (CMV) [46], and other viruses in vitro and in vivo (Figure 1). Fullerenes and their derivatives, as a class of new, broad-spectrum antiviral drugs, have attracted increasing attention as a potential treatment for SARS-CoV-2.

Because fullerenes are insoluble in water and polar media, their use in biomedicine is extremely complicated [47]. To increase biocompatibility, the cage structure of fullerenes needs to be modified with appropriate hydrophilic functional groups. The modified structure and properties of the carbon cage may facilitate new applications in different biological systems. Because the fullerene carbon cage has multiple modifiable reaction sites, many fullerene derivatives with well-defined structures have been synthesized using regioselective functional group derivatization strategies. Therefore, fullerenes serve as ideal scaffolds for different bioactive drugs. Studies on the synthesis and antiviral activity of fullerenes and their derivatives have facilitated a deeper understanding of the relationship between fullerene structure and bioactivity.

In the past, hundreds of fullerene derivatives have been synthesized and used to inhibit viruses in vitro. Most of these derivatives are water soluble. These fullerene derivatives can be classified as the following six types: (1) amino acid, peptide, and primary amine derivatives; (2) piperazine and pyrrolidine derivatives; (3) carboxyl derivatives; (4) hydroxyl derivatives; (5) glycofullerene derivatives; and (6) fullerene complexes. Numerous antiviral studies have been conducted to evaluate fullerene C_60_ and its derivatives; this review assesses the latest research on the ability of fullerene C_60_ and its derivatives to inhibit virus replication.

## 2. Synthesis and Antiviral Properties of Amino Acid, Peptide, and Primary Amine Fullerene Derivatives

HIV-1-specific protease (HIVP) is an effective target for antiviral therapy. Its active site can be roughly described as an open-ended cylindrical cavity composed almost entirely of hydrophobic amino acids [48]. Since spherical C_60_ derivatives are hydrophobic and have a similar radius as the cylindrical cavity, strong hydrophobic interactions may occur between the active site surface and C_60_ derivatives. Therefore, C_60_ derivatives are potential inhibitors of HIVP.

In 1993, Friedman et al. [39] reported a landmark study based on model building and experimental evidence. Theoretical calculations revealed that the hydrophobic cavity of HIVP accommodates C_60_ molecules; the spherical C_60_ molecules fit perfectly within the active site, facilitating strong interactions between HIVP and fullerene. However, because C_60_ spheres are insoluble in polar solvents, it is critical to dissolve C_60_ in a medium suitable for biological testing. By modifying fullerene with strong polar groups, Sijbesma et al. [49] synthesized the water-soluble fullerene derivative bis (phenylenaminosuccinic acid)-C_60_ (**1**) (Figure 2). In vitro studies revealed that **1** inhibited acute and chronic HIV-1 infection in human peripheral blood mononuclear (PBM) cells, with a half-effective concentration (EC_50_) of 7.0 µM, while showing no cytotoxicity to uninfected PBM cells. This work is of great significance because fullerene derivatives as virus inhibitors are unprecedented in the field of antiviral research.

According to the theoretical analysis, the main driving force behind the binding between HIVP and fullerene derivatives is the hydrophobic interaction between the non-polar active site surface of the protease and the non-polar fullerene surface [50,51]. In order to improve antiviral activity, Friedman et al. proposed the addition of appropriate functional groups at specific positions on the fullerene derivatives [39] that would interact with the protease, generate electrostatic and/or hydrogen bonds as well as van der Waals forces, and subsequently increase the binding constant by several orders of magnitude. Therefore, compound **2** with two amino groups (Figure 3) was designed as an ideal “second generation” fullerene derivative. In addition to forming van der Waals forces with the non-polar HIVP surface, the cationic sites on the fullerene surface can form salt bridges with the catalytic aspartates on the floor of the active site, thus improving the binding strength. Although compound **2** was an ideal model of a fullerene derivative, the pure isomer was difficult to obtain, owing to a lack of regioselectivity. Subsequently, Prato and co-workers [52] proposed the design, simulation, and synthesis of the C_60_ diamine derivative **3**, which was similar to **2**. In the PM3-minimized structure of **3**, the N–N distance between the two amino groups is 0.51 nm, while that in **2** is 0.55 nm, suggesting that the spatial arrangement of the two amino groups in **3** is very similar to that in **2**.

In order to determine the binding energy between **3** and HIVP, Prato and co-workers [52] conducted a simulation using the Discover (Biosym/MSI) program with a *cvff* force field. Figure 4a depicts **3** binding to the cavity region of HIVP to form a complex. Compared with unmodified fullerene C_60_, the complex formed by the binding of **3** to HIVP exhibited significant improvements. When fullerene was protonated with a monoamino group, the relative binding energy was approximately −134 kJ/mol, and when it was protonated with a diamino group, the relative binding energy was approximately −105 kJ/mol. The greater binding energy of compound **3**, with a single amino group, is due to the hydrogen bond interactions between the neutral –NH_2_ and –COOH groups of neutral aspartic acid, rather than the hydrogen bond interactions between the –NH_3_^+^ group and –COOH. Figure 4b illustrates the active site of the complex, highlighting the hydrogen bonds between the HIVP cavity and **3**. The interatomic distance between the N atom in each amine/ammonium molecule and the O atom in the carboxyl/carboxylic acid is about 0.28 nm. This strong interaction suggests the contribution of Coulombic attraction.

In the same time period, many research groups synthesized various fullerene derivatives with different functional groups and demonstrated effective virus inhibition by introducing appropriate carboxylic acid and amino acid groups at specific positions on fullerenes. Among them, the fullerene dendritic amino acid derivative **4**, prepared by Brettreich and Hirsch [53] (Figure 5), is highly water soluble and has an EC_50_ of 0.22 µM in human lymphocytes acutely infected with HIV-1 [54]. This compound also shows no toxicity up to 100 μM in PBM, Vero, and CEM cells. The fullerene amino acid derivative **5** (Figure 5), synthesized by Mashino et al. [40], strongly inhibits HIV reverse transcriptase (HIV-RT), with an EC_50_ of 0.029 μM. Additionally, Toniolo et al. [55] synthesized fullerene peptide derivatives, which also exhibit anti-HIVP activity.

In 2012, Troshin and coworkers [43] reported that multi-functional C_60_ amine and amino acid derivatives could be readily prepared from hexachlorofullerene C_60_Cl_6_ (**6**). The synthesized fullerene derivative contains at least five hydrophilic functional groups (such as amino or carboxyl groups). As shown in Figure 6, a fullerene amino acid ester (**8**) can be obtained via the reaction of C_60_Cl_6_ with an amino acid ester (**7**). The amino acid ester groups in compound **8** can be hydrolyzed to obtain fullerene amino acid derivatives under acidic conditions. In order to further increase the water solubility of fullerene amino acid derivatives, fullerene amino acid potassium salts are formed by adding potassium carbonate.

In vitro cell experiments [43] showed that the carboxylic acid potassium salt **8** has low cytotoxicity to HSV-sensitive Vero cells (CC_50_ > 1.3 mM) and human CMV-sensitive human embryonic lung fibroblasts (CC_50_ > 0.5 mM). Meanwhile, compound **8** showed pronounced antiviral activity, with an EC_50_ of 0.26 µM for HSV and 37.6 µM for CMV. Combining CC_50_ and EC_50_ values, the selectivity indices of compound **8** for HSV and CMV are >5000 and 14, respectively, indicating that compound **8** has potential as a new antiviral drug against HSV and CMV.

## 3. Synthesis and Antiviral Studies of Fullerene Piperazine and Pyrrolidine Derivatives

As shown in Figure 7, the reaction of C_60_Cl_6_ with *N*-methylpiperazine (**9**) can efficiently generate a fullerene-piperazine derivative (**10**) in the absence of any base [43]. By adding six times the amount of compound **9** to a C_60_Cl_6_ toluene solution, **10** is precipitated immediately, with more than a 95% yield. In vivo cell experiments have indicated that **10** exhibits high acute toxicity when administered via intraperitoneal injection in mice, while **8** shows very low acute toxicity, suggesting the latter is safe for biomedical applications. This example also revealed that the toxicity of the water-soluble fullerene derivatives depends largely on the organic functional groups attached to the fullerene carbon cage.

According to the literature [51], a regioisomer mixture of C_60_ pyrrolidine derivatives exhibits good anti-HIV-1 activity. The pyrrolidine derivative has a low EC_50_ and exhibits little toxicity to Vero and PBM cells. However, a relationship between the structure and activity of the reported isomer mixture of fullerene pyrrolidine derivatives has not been established. In order to better understand which structural characteristics can be modified to improve anti-HIV activity, Prato and coworkers [56] synthesized a series of new, pure fullerene pyrrolidine derivatives through the Prato reaction (Figure 8).

The Bis-adduct fullerene pyrrolidine derivative has eight regioisomers. The yields of the eight isomers differ; *cis*-Bis-adducts are produced at extremely low yields and *trans*- and *equatorial*-adducts are produced at relatively high yields. Hence, Prato and coworkers were able to separate *trans-* and *equatorial*-isomers (**12**–**15** and **16**–**19**, respectively) and study their structure and activity. Additionally, a Mono-adduct (**11**) and other Bis-adducts (**20**–**21**, **22**–**23**) were synthesized (Figure 8).

The anti-HIV activity and cytotoxicity of all fullerene pyrrolidine derivatives (**11**–**23**) was tested in lymphocyte (CEM) cultures infected with HIV-1 or HIV-2. As shown in Table 1, mono-functionalized derivative **11** and Bis-adduct derivatives **12**–**15**, **20**, **21** and **23** showed low anti-HIV-1 activity, while the corresponding quaternary ammonium pyrrolidine derivatives **16**–**19** showed high anti-HIV-1 activity at low concentrations (EC_50_: 0.40–2.60 µM). These results suggest that the inhibitory effect on HIV-1 might be related to electrostatic interactions.

The biological activity of pyrrolidine derivatives of fullerene varies among the regioisomers. As shown in Table 1, the anti-HIV-1 activity of the *trans*-2 isomer (**16**) was 2.4~6.5 times that of the corresponding *trans*-3 (**17**), *trans*-4 (**18**), and equatorial (**19**) isomers. The *trans*-2-tetraacetic acid derivative (**22**) showed high anti-HIV-1 activity at low concentrations, while the equatorial isomer (**23**) showed low activity. The introduction of malonate resulted in the loss of anti-HIV-1 activity of pyrrolidine derivatives **20**–**21**. Most synthetic fullerene pyrrolidine derivatives exhibit pronounced toxicity in human CEM cells (CC_50_: 3.02~13.2 µM), but monofunctional derivative **11** exhibit relatively low toxicity (CC_50_: 44 µM). This toxicity most likely results from the strong amphiphilic properties of these pyrrolidine derivatives, which can cause the rupture of the cell membrane and subsequent cell death.

To define the structure-activity relationship, Prato and co-workers [57] prepared Bis-adduct fullerene pyrrolidine derivatives (**24**–**28**) through a [3+2] dipolar cycloaddition reaction between azomethine ylides and C_60_ (Figure 9) and subsequently studied their anti-HIV-1 and anti-HIV-2 activity and cytotoxicity in CEM culture. As shown in Table 2, the anti-HIV-1 activity of the trans-isomers (**24**–**26**) is about 2–10 times that of the corresponding *cis*-3 isomer (**28**), while the equatorial-isomer (**27**) exhibits no antiviral activity against HIV-1. Among the *trans*-isomers, *trans*-2 (**24**) and *trans*-3 (**25**) demonstrate higher levels of anti-HIV-1 (EC_50_ = 0.21 and 0.35 µM, respectively) and anti-HIV-2 activity (EC_50_ = 0.21–1.0 µM). The anti-HIV-1 activity of the *trans*-4 isomer (**26**) is significantly lower than that of compounds **24** and **25** (EC_50_ = 1.08 µM), and the anti-HIV-2 activity is also lower (EC_50_ = 2.5 µM), which is consistent with the experimental results for fullerene pyrrolidine derivatives **11**–**23**. Fullerene pyrrolidine derivatives **24**–**28** also exhibit toxicity in human CEM cells (CC_50_ = 2.9–28.7 µM). The CC_50_/EC_50_ ratio of the *trans*-3 isomer (**25**) is 26, higher than that of the reference compound **16** (CC_50_/EC_50_ = 12).

As mentioned, Mashino et al. (2005) reported that fullerene amino acid-type derivatives exhibited HIV-RT inhibitory activity, while cationic fullerene derivatives such as the pyrrolidinium-type derivatives showed weaker HIV-RT inhibitory activity [40]. The carboxyl groups on the pyrrolidine-type fullerene derivatives were considered crucial to HIV-RT inhibitory activity [40,58]. However, ten years later, Mashino et al. found that fullerene pyrrolidine-pyridine and pyrrolidine-pyridinium salt derivatives without any carboxyl groups [42], such as **29**–**40**, which are functionalized with pyridine or pyridinium groups (Figure 10), exhibit strong HIV-RT inhibition. This is useful information for the future design of fullerene derivatives as HIV-RT inhibitors. In addition to HIV-RT inhibition, recently, Kobayashi et al. [59] found that compound **34** can inhibit HIV-PR, and HCV NS5B polymerase (HCV NS5B) with IC_50_ values in the micromolar range. Compound **41**, the exo-substituent on the most potent derivative (**3****4**), exhibits no HIV-RT inhibitory activity in cell culture, indicating that HIV-RT inhibition is dependent on the fullerene skeleton. The conventional trypan blue dye exclusion test (Table 3) was used to evaluate the cytotoxicity of all fullerene pyrrolidine-pyridine and pyrrolidine-pyridinium salt derivatives (**29**–**40**) to HL60 cells. The CC_50_ of all derivatives except **35** (CC_50_ = 39.4 µM) was greater than 50 µM, which suggests that the new fullerene-pyrrolidine-pyridine or pyridinium salt derivatives effectively inhibit HIV-RT activity without damaging living cells.

In 2016, Echegoyen and co-workers [41] reported a novel cationic *N*,*N*-dimethyl C_70_ fullerene-pyrrolidine iodized salt derivative (**42**–**44**), with fullerene C_70_ as the starting material (Figure 11), that inhibits more than 99% of HIV-1 infectivity at a low micromolar concentration. These three compounds have an EC_50_ of 0.41, 0.33, and 0.54 µM, respectively. An analysis of the life cycle of HIV-1 suggested that these compounds inhibit viral maturation by influencing the processing of Gag and GAG-POL. Significantly, fullerene derivatives **42**–**44** do not inhibit protease activity in vitro, and strongly interact with immature HIV capsid proteins in a pull-down assay. Moreover, these compounds may block infection by viruses carrying either a mutant protease that is resistant to multiple protease inhibitors, or the mutant Gag protein, which is resistant to the mature inhibitor bevirimat. Fullerenes **42**–**44** do not inhibit HIV-1 proteases at doses that strongly inhibit HIV-1 infection in vitro, suggesting that this mechanism is independent of the HIV-1 protease. This finding differed from previous reports that fullerene derivatives affect HIV-1 protease activity in vitro. Echegoyen et al. [60] then proposed that fullerene derivatives **42**–**44** act through a novel anti-HIV-1 mechanism, rather than interacting with other capsid proteins as previously reported. Unraveling the details of this mechanism will facilitate the discovery of novel anti-HIV-1 inhibitors.

Additionally, Tollas et al. [44] reported that fullerene pyrrolidine derivatives functionalized with hydrophilic sugar groups. While these derivatives have no inhibitory effect on influenza virus hemagglutinin, they exhibit a good inhibitory effect on neuraminidase.

## 4. Synthesis and Antiviral Studies of Fullerene Carboxyl Derivatives

As mentioned, hydrophilic functional groups such as amino or amino acid groups can be directly or indirectly attached to the skeleton of fullerenes to increase water solubility. These water-soluble fullerene derivatives have exhibited high levels of antiviral activity. Instead of amino or amino acid groups, a single carboxylic acid group can be used to modify fullerenes to increase their water solubility and improve their antiviral activity. In 1996, Schuster et al. [51] synthesized 11 new water-soluble fullerene derivatives, among which fullerene carboxylic acid derivatives **45** and **46** (Figure 12) exhibit antiviral activity against HIV-1 at low micromolar concentrations, with an IC_50_ of 2.2 and 6.3 μM, respectively.

In 2007, Troshin and coworkers [61] proposed an effective method for the synthesis of water-soluble fullerene carboxylic acid derivatives. With C_60_Cl_6_ as the starting point, C_60_(Ar)_5_Cl, with ester groups linked to aryl groups, was obtained via the simple and efficient Friedel–Crafts arylation of C_60_Cl_6_ (**6**) with methyl esters of phenylacetate at 100 °C. The fullerene carboxylic acid derivative (**48**) was prepared in almost a quantitative yield by removing the methyl group from the methyl ester under acidic conditions, as shown in Figure 13. Compound **48**, with five carboxyl groups, is insoluble in water but soluble in DMSO. In order to improve the water solubility, potassium carbonate was added to neutralize the carboxylic acid group of **48** and form the corresponding ionic potassium salts, with a water solubility of up to 50–100 mg/mL at pH < 7.5. A virus-induced cytopathicity assay revealed that the fullerene carboxylic acid potassium derivative has pronounced anti-HIV-1 activity, with an IC_50_ of 1.20 ± 0.44 μM and a low cytotoxicity (>52 μM).

Usually, only five chlorine atoms in **6** are replaced; the sixth chlorine atom cannot be substituted due to steric hindrance. Recently, Troshin and coworkers [62] found that the sixth chlorine atom can be substituted by an alkyl group through a reverse Arbuzov reaction between trialkyl phosphite (P(OR)_3_) and the fullerene derivative C_60_(Ar)_5_Cl (Figure 14). More significantly, the introduction of different R groups through the reverse Arbuzov reaction affects the antiviral activity of the carboxyl fullerene derivatives, establishing a fundamental correlation between the structure of carboxyl fullerene derivatives and their antiviral activity. Experiments on the inhibition of the influenza H3N2 virus showed that compounds **49** and **50** (R = Et and Me, respectively) were quite active, while the fullerene derivative **48** (R = Cl) was completely inactive. Specifically, **49** and **50** effectively inhibited influenza virus H3N2 at nanomolar concentrations, with an EC_50_ of 500 nM and 100 nM, respectively; both compounds were more active against H3N2 than the commercial drugs zanamivir (EC_50_ = 3.0 μM) and amantadine (EC_50_ = 1.3 μM). However, there were no significant differences in the HIV-1 inhibitory activity of the fullerene derivative C_60_(Ar)_5_Cl after the alkyl substitution of chlorine atoms.

Recently, Troshin and coworkers [63] synthesized the fullerene carboxylic acid derivative C_70_[*p*-C_6_H_4_(CH_2_)_n_COOH]_8_ by Friedel–Crafts arylation of chlorofullerene C_70_Cl_8_ with unprotected carboxylic acids. The obtained carboxylic acid fullerene C_70_ derivatives showed significant antiviral effects against HIV and the influenza viruses H1N1 and H3N2. The EC_50_ value of anti-HIV-1/NL4.3 (X4) is close to 1.0 µM, which indicates that this derivative is more effective against the virus than the commercial drug tenofovir. Additionally, Troshin and coworkers synthesized a variety of carboxylic acid fullerene derivatives, such as the carboxylic acid thiofullerene derivative, **51** [64]; polycarboxylic acid fullerene derivative, **52** [65]; and tetracarboxylic acid methanofullerene derivative, **53** (Figure 15) [66]. All these compounds exhibit inhibitory effects against HIV-1, HIV-2, influenza A (H3N2), HSV, and CMV, with low toxicity.

Modern antiviral drugs have extended patients’ lives and improved their quality of life, but unsolved problems remain, such as toxicity, limited bioavailability, and drug resistance (rapid cyclic changes in influenza strains reduce the effectiveness of commonly used vaccines). Therefore, finding new antiviral drugs has become an urgent problem. The research by Troshin and coworkers on carboxylic acid fullerene derivatives has provided more possibilities for the development of new and effective antiviral drugs [61,62,63,64,65,66].

## 5. Synthesis and Antiviral Studies of Fullerene Hydroxyl Derivatives

Fullerenol is a polyhydroxylated fullerene C_60_ derivative with suitable water solubility and biocompatibility. In 2013, Eropkin et al. [67] synthesized a series of fullerenols (Figure 16). Three different groups of fullerenols, C_60_(OH)_12–14_, C_60_(OH)_18–24_, and C_60_(OH)_30–38_, can be prepared by changing the reaction conditions to control the number of hydroxyl groups attached to the fullerene skeleton. Experimental studies revealed that fullerenols containing 12~14 hydroxyl groups are insoluble in water and have no biological activity when introduced into cell culture as suspensions. The other two groups of fullerenols show broad spectrum antiviral activity in vitro against the human influenza viruses H1N1 and H3N2, avian influenza virus A (H5N1), adenovirus, human HSV, and respiratory syncytial virus. C_60_(OH)_18–24_ demonstrates better antiviral activity than C_60_(OH)_30–38_. Moreover, the three water-soluble fullerenols exhibit no toxicity in vitro to human and animal cells.

Compared with other fullerene derivatives, fullerenols are relatively easy to prepare, requiring only one organic reaction step. So far, only one example of polyhydroxyfullerene (C_60_(OH)_8_) with a well-defined structure has been reported, by Gan and co-workers [68]. However, the insufficient number of hydroxyl groups in C_60_(OH)_8_ limits its biological applications. At present, even by purification via high performance liquid chromatography, it is impossible to obtain pure regioisomer fullerenols from the mixture of polyhydroxyfullerenes. Therefore, the fullerenols used in current biological studies have been a mixture of regioisomers, which restricts the potential of fullerenols as standard drug candidate molecules.

## 6. Synthesis and Antiviral Studies of Glycofullerene Derivatives

Carbohydrate and protein interactions dominate many biological processes, including inflammation, embryogenesis, tumor development and metastasis, and pathogen infection [69]. These interactions are characterized by high selectivity, metal ion dependence, and compensation for low affinity through multivalent interactions [70]. Finding a suitable system to realize the polyvalent expression of sugars has been a subject of extensive research [71,72]. Due to the lack of information on the proper orientation of the ligands required to obtain the strongest interactions, chemists have experimented with all possible scaffolds. Calixarene [73], gold nanoparticles [74], polymers [75], liposomes [76], dendritic macromolecules [77], and fullerenes [78] are most commonly used as scaffolds to construct multivalent glycoconjugates. The advantages associated with fullerenes over other nanostructures are their three-dimensional (3D) structure and the ability to functionalize different positions of the C_60_ cage in a controlled way [79]. In this sense, fullerenes can be thought of as a special class of spherical scaffolds ideal for building a multivalent spherical ligand. Martin and coworkers [78] proposed that spherical carbohydrate derivatives of fullerenes, with fullerenes used as scaffolds, could serve as an interesting sugar analog. The rigid spherical scaffold allows distance to be maintained between the two ligands (the diameter of fullerenes is 1 nm, plus the distance provided by the dendritic moiety), thus increasing the chances of obtaining effective multivalent interactions. In addition, due to the symmetry of the multivalent system, the 3D orientation of these ligands at 360° better mimics the surface of a pathogen, such as HIV; thus, the molecule is more likely to encounter a receptor.

Based on an octahedral addition pattern, soluble hexakis-adduct glycofullerene derivatives can act as spherical carbohydrates, thus serving as a potential multivalent spherical ligand. In 2013, Martin and coworkers [79] designed and synthesized a class of hexakis-adduct glycofullerene derivatives (**54**), containing 36 mannoses (Figure 17), and used them to inhibit cell infection by pseudotyped Ebola virus particles. This was the first time that glycofullerene derivatives were demonstrated to effectively inhibit cell infection. In the pseudotyped Ebola infection model, the antiviral activity of **54** was in the low nanomolar range, with an IC_50_ of 0.3 μM. The glycofullerene with 12 galactosyl had no inhibitory effect on virus infection, indicating that the inhibitory effect is dependent on mannose. Interestingly, only an increase in the valence of glycofullerene resulted in a loss of the antiviral effect. This phenomenon is related to the spatial crowding of sugars at the surface of glycofullerene. Martin et al. speculated that the high binding affinity occurs not only because of the extensive spatial presentation of multivalent ligands, but also the frequent interactions between the ligands and corresponding receptors. They demonstrated that a rational design of compounds with the same valency but longer spacers can significantly increase the antiviral activity, likely due to more efficient interactions with receptors. Therefore, the selection of suitable scaffolds (such as spherical fullerenes) to achieve multivalence, as well as the accessibility and flexibility of ligands, are key factors for improving antiviral activity.

The aforementioned studies have demonstrated that using multivalent glycofullerenes to block lectin receptors on the cell surface is a promising method for inhibiting virus entry into cells. However, creating large enough multivalent glycofullerenes to improve the binding ability between ligands and virus receptors remains a challenge. In 2015, Martin and coworkers [45] conducted an impressive study of water-soluble fullerenes. They synthesized three water-soluble glycofullerene derivatives (**56**–**58**) with 12 fullerene spheres modified with 120 sugars, starting from the hex-adduct fullerene derivative (**55**), namely, the “super sphere” (Figure 18). In addition to the core fullerene sphere, the other 12 fullerenes in the super sphere contain 10 sugars each, totaling 120 sugars, and have diameters up to 4 nm.

The mannose in compounds **56**–**58** is critical in blocking virus entry into cells. As expected, **57**, with 120 galactosyl species, did not inhibit the infection process. Glycofullerene derivatives **56** and **58**, with 120 mannoses, exhibited high levels of antiviral activity in the pM~nM range. As shown in Table 4, **56** can effectively block Ebola virus infection in the nM range, with an IC_50_ of 20.4 nM. Compound **58** is almost 10 times more potent in the antiviral infection process, with an IC_50_ of 667 pM. Fullerene **54**, linked with 36 mannoses, produced relative inhibitory effect (RIP) values of at least two orders of magnitude smaller than those of **56** and **58**. Compared to previously reported results, **56** and **58** are the most effective compounds against Ebola virus infection in vitro. Compounds **56**–**58** showed no appreciable cell cytotoxicity at the concentrations used in the virus inhibition experiments. Owing to their biocompatibility and spherical structure, fullerenes have become ideal scaffolds for studying multivalent interactions. However, further research is needed to determine whether the obtained glycofullerene derivatives can be used for practical applications.

In order to enhance multivalency and improve the biocompatibility of glycofullerene derivatives, Martin and coworkers [80] (2019) synthesized tridecafullerene derivatives containing up to 360 1,2-mannobiosides via the strain-promoted azide–alkyne cycloaddition method. The obtained glycofullerene derivative showed pronounced antiviral activity against Zika virus and dengue virus, with an IC_50_ of 67 and 35 pM, respectively.

Additionally, Martin and coworkers [81] synthesized a series of amphiphilic glycodendrofullerene [60] monoadducts (**59** and **60**) through the “click chemistry” reaction. In aqueous media, the glycodendrofullerenes can self-assemble into large, compact micelles with a uniform size and spherical shape. Antiviral tests showed that these aggregates of **59** and **60** can effectively inhibit Ebola virus infection in the nM range, with an IC_50_ of 424 nM and 196 nM, respectively. However, these compounds are inferior to **56** and **68**.

Ebola virus has a filament-like structure and is similar to single-walled carbon nanotubes (SWCNTs) and multi-walled carbon nanotubes (MWCNTs) in shape. SWCNTs are usually 1.5 to 3.0 nm in diameter. They match Ebola virus in length, at 20 to 1000 nm. On the other hand, HIV has a roughly spherical shape of about 120 nm in diameter and is very similar in shape and size to spherical fullerenes and single-walled nanocones (SWCNHs) [82]. Therefore, Martin and coworkers proposed that SWCNTs, MWCNTS, and SWCNHs could be used as virus-mimicking nanocarbon platforms, and after chemical modification, could interact with the receptors in a multivalent manner. To employ the novel strategy for designing antiviral agents against HIV or Ebola virus (designing glycofullerene and nanocarbon complexes that mimic the virus surface and interfere with the infection of receptors on the corresponding cell surface), Martin et al. [83] (2018) used the “click chemistry” reaction to covalently connect glycofullerene to SWCNTs, MWCNTs, or SWCNHs. The multivalent hybrid glycoconjugate consisting of MWCNTs and glycofullerene (**61**) is shown in Figure 19. After chemical modification, the water solubility of the MWCNTs was significantly improved. In tests of efficiency in blocking artificial Ebola virus infection with three types of multivalent hybrid glycoconjugates, **61**, based on MWCNTs and fullerene functionalization, was the most effective inhibitor of viral infection, with an IC_50_ of 0.37 µg/mL.

Cyclodextrins (CDs) are a series of cyclic oligosaccharides containing 6–12 D-glucopyranose units. They have a slightly conical, hollow, cylindrical three-dimensional ring structure and have been widely studied because of their significant solubility in water. The CD molecules containing 6, 7, and 8 glucose units are called α-, β- and γ-CD, respectively. In 2012, Zhang and coworkers [84] prepared a water-soluble α-CD-C_60_ conjugate (**62**), in which C_60_ and α-CD were linked by two flexible alkyl chains (Figure 20). Interestingly, this water-soluble compound showed pronounced anti-HCV entry activity, with an IC_50_ of 0.17 µM.

In 2018, Zhang and coworkers [85] continued to design and synthesize seven *α*-CD-C_60_ copolymers and two γ-CD-C_60_ copolymers [85]. However, the newly obtained *α*-CD-C_60_ conjugates did not exhibit inhibitory activity against HCV. Subsequently, all nine CD-C_60_ conjugates were assessed in terms of their activity against the influenza virus H1N1. No conjugates exhibited cytotoxicity at 100.0 µM. The two *γ*-Cd-C_60_ conjugates demonstrated higher anti-H1N1 activity, with IC_50_ values of 87.7 µM and 75.0 µM, respectively. Because they exhibit less aggregation in aqueous solutions, the two *γ*-Cd-C_60_ conjugates are the most water soluble of the nine conjugates. This trait might be related to their higher inhibitory efficiency against H1N1.

## 7. Synthesis and Antiviral Studies of Fullerene Complexes

As mentioned, to enhance the water solubility of pristine fullerenes, various hydrophilic functional groups (amino acids, sugars, or calixarene) can be directly or indirectly used to modify fullerenes through covalent bonds. Alternatively, pristine fullerenes can be dispersed in polyvinylpyrrolidone (PVP), Triton X-100, dioctadecyldimethylammonium bromide, or lecithin to form complexes.

Sirotkin et al. [86] added an aqueous solution of the C_60_/PVP complex to a suspension of H1N1 particles. As a result, the number of virus particles with a broken lipoprotein envelope increased dramatically, possibly due to the fusion of the C_60_/PVP complex with the influenza virus.

Yang and coworkers [87] prepared a fullerene [60] liposome complex (Figure 21) and studied its anti-H1N1 activity in vivo. The fullerene liposome complex significantly reduces the average lung virus yields and lung index; prolongs the mean time to death; and decreases the mortality of mice infected with H1N1. In addition, the fullerene liposome complex has good water solubility and low toxicity, and its anti-influenza activity in vivo is much higher than that of rimantadine. Therefore, the fullerene [60] liposome complex is a promising clinical candidate drug against influenza infection.

## 8. Conclusions

This review summarized the latest antiviral research conducted on fullerenes and their derivatives. Numerous water-soluble fullerene derivatives or fullerene complexes have shown great antiviral potential, mainly because fullerenes have three advantages. First, pristine fullerenes are hydrophobic, which is conducive to the formation of strong hydrophobic interactions with the active site surfaces of viruses. Second, hydrophilic groups with various functions (such as amino, carboxyl, amino acid, hydroxyl, pyrrolidine, and sugar groups) can be used to selectively modify the unique spherical skeleton of fullerenes via organic reactions. Third, fullerenes and their derivatives exhibit no or low cytotoxicity at relatively high concentrations. Although fullerenes are promising prospective antiviral drugs, antiviral research on fullerenes requires improvement. Most fullerene derivatives exhibit good antiviral effects in vitro, but the antiviral mechanism has not been thoroughly studied. Additionally, most of the studies on fullerenes have only focused on virus inhibition in vitro; there have been few antiviral studies in vivo, and relevant clinical studies involving fullerenes have not been conducted.

Viruses constantly threaten human health. Fullerenes have become an important molecular platform for the development of antiviral drugs. Research on fullerenes as antiviral drugs urgently needs the joint efforts of scientists working in synthesis, molecular design, biology, and medicine. Some fullerene derivatives display inhibitory activity against multiple types of viruses. Therefore, fullerene derivatives have the potential to become a class of broad-spectrum antiviral drugs effective against SARS-CoV-2, which remains a global threat. We believe that this review will encourage more researchers to synthesize fullerene derivatives and study their antiviral properties and applications.

## Figures and Tables

**Figure 1 nanomaterials-12-02547-f001:**
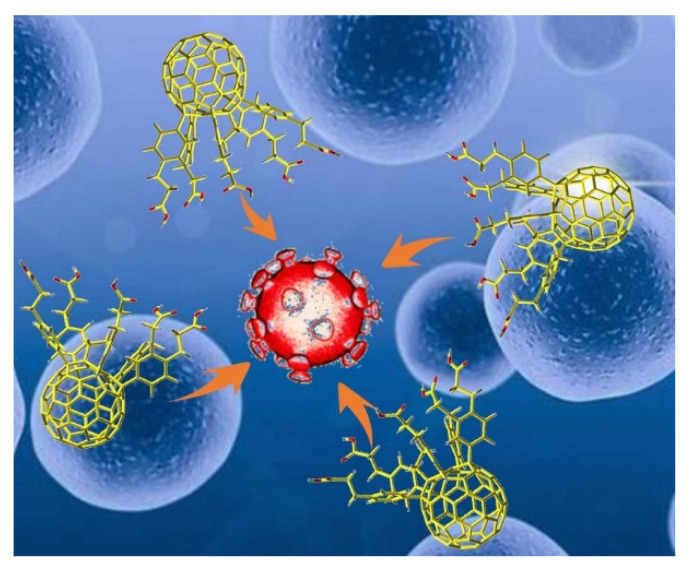
Possible interaction between fullerene molecules and coronavirus, in which fullerene molecules inhibit virus replication.

**Figure 2 nanomaterials-12-02547-f002:**
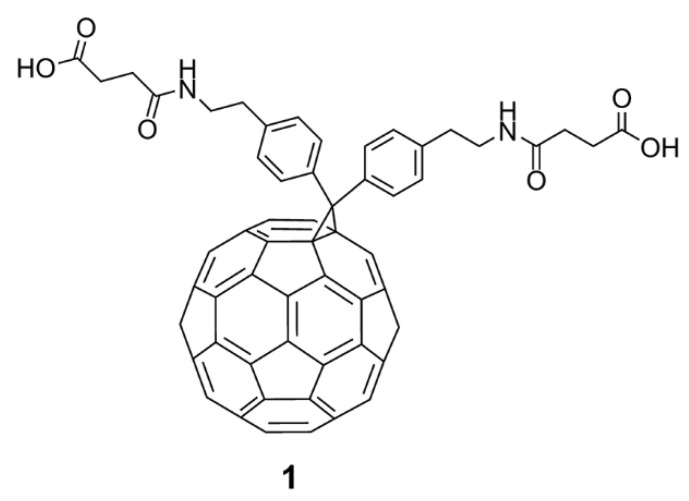
The first example of a water-soluble fullerene derivative (**1**) used as a virus inhibitor.

**Figure 3 nanomaterials-12-02547-f003:**
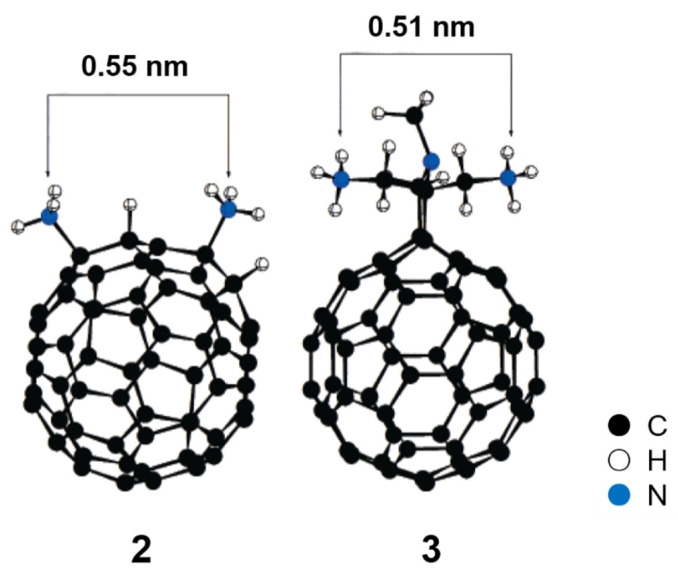
PM3-minimized structures of compounds **2** and **3**. Adapted with permission from Ref. [52]. Copyright 2000 American Chemical Society.

**Figure 4 nanomaterials-12-02547-f004:**
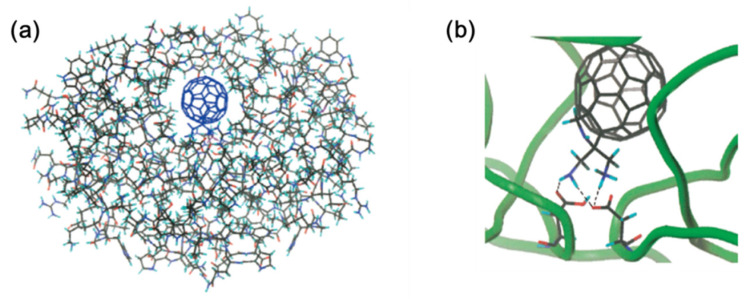
Computer-simulated interactions between compound **3** and HIVP. (**a**) Accommodation of compound **3** in the cavity of HIVP. (**b**) Closer view of the complex of compound **3** and HIVP. Adapted with permission from Ref. [52]. Copyright 2000 American Chemical Society.

**Figure 5 nanomaterials-12-02547-f005:**
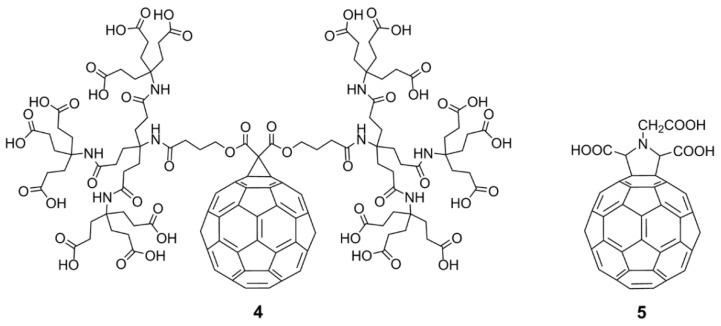
Fullerene amino acid derivatives (**4**) and (**5**) as potential inhibitors of HIV-1 cell replication.

**Figure 6 nanomaterials-12-02547-f006:**
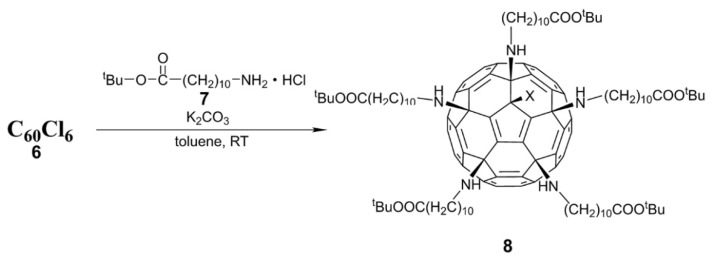
Synthesis route of fullerene amino ester derivative (**8**).

**Figure 7 nanomaterials-12-02547-f007:**
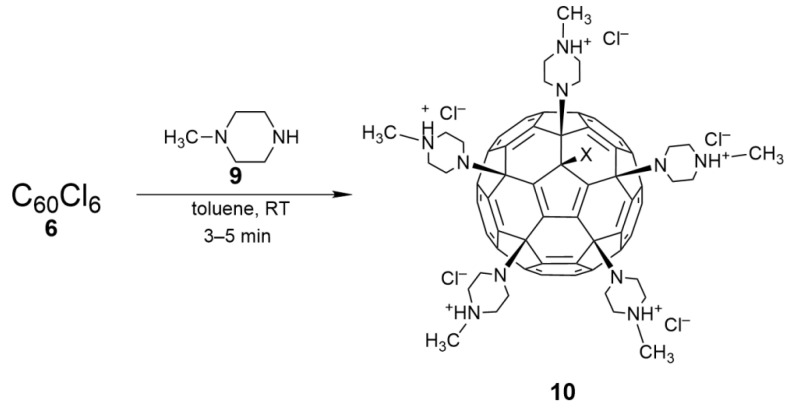
Synthesis route for fullerene penta-*N*-methyl piperazine salt (**10**).

**Figure 8 nanomaterials-12-02547-f008:**
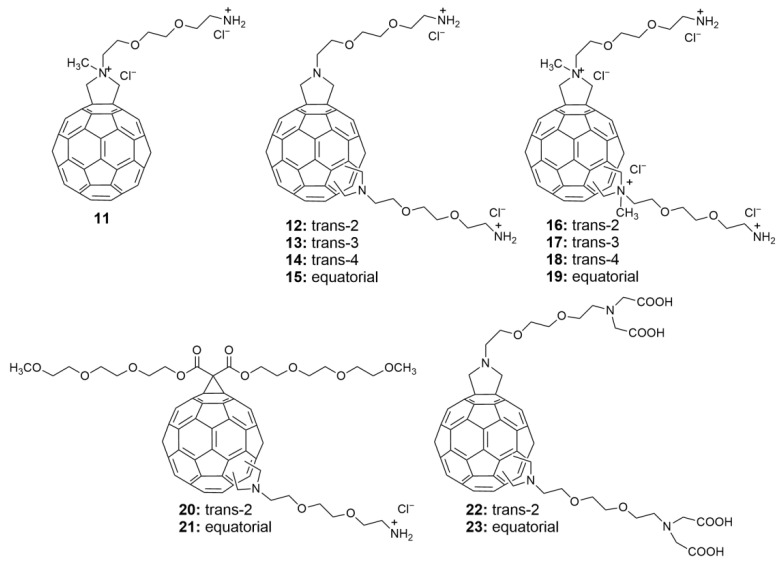
Structure diagram of fullerene pyrrolidine derivatives (**11**–**23**).

**Figure 9 nanomaterials-12-02547-f009:**
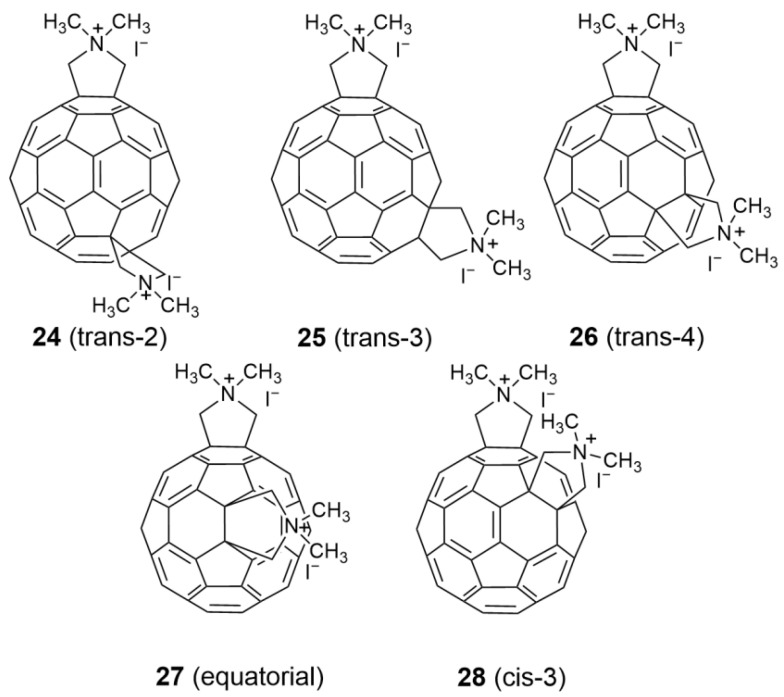
Structure diagram of fullerene pyrrolidine derivatives (**24**–**28**).

**Figure 10 nanomaterials-12-02547-f010:**
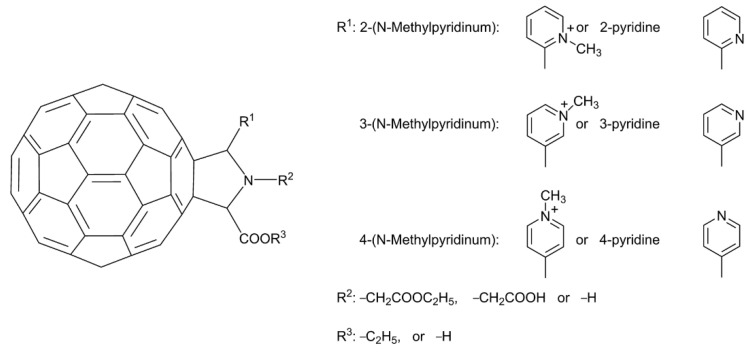
Structure diagram of fullerene pyrrolidine derivatives (**29**–**41**).

**Figure 11 nanomaterials-12-02547-f011:**
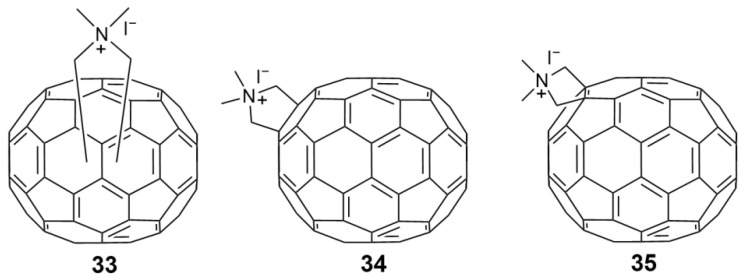
Chemical structures of the cationic *N*,*N*-dimethyl [70]Fullerene pyrrolidine iodide derivatives (**42**–**44**).

**Figure 12 nanomaterials-12-02547-f012:**
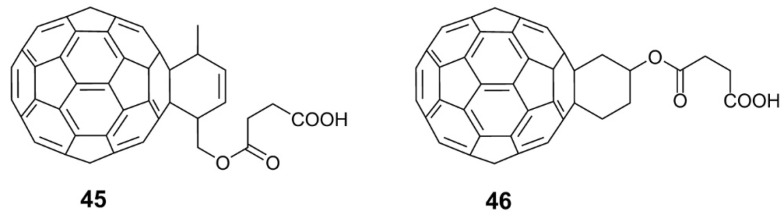
Structural diagram of carboxylic fullerene derivatives (**45**–**46**).

**Figure 13 nanomaterials-12-02547-f013:**
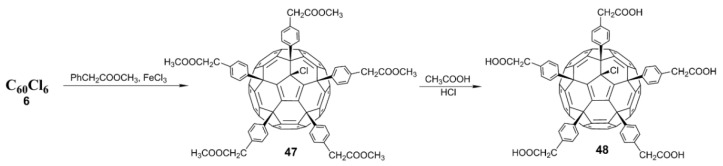
Schematic diagram of synthetic route of carboxylic fullerene derivative (**48**).

**Figure 14 nanomaterials-12-02547-f014:**
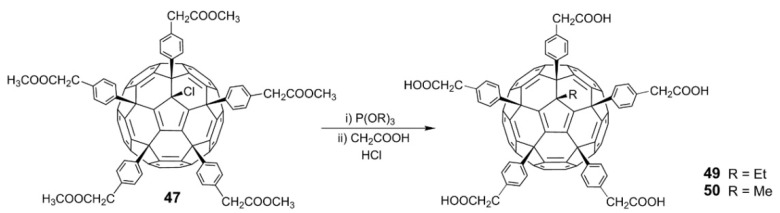
Schematic diagram of synthetic routes of carboxylic fullerene derivatives **49** and **50**.

**Figure 15 nanomaterials-12-02547-f015:**
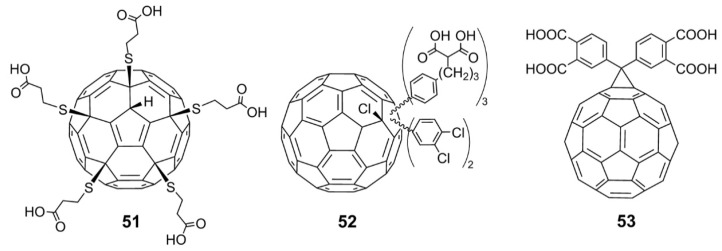
Structure diagram of polycarboxylic fullerene acid derivatives (**51**–**53**).

**Figure 16 nanomaterials-12-02547-f016:**
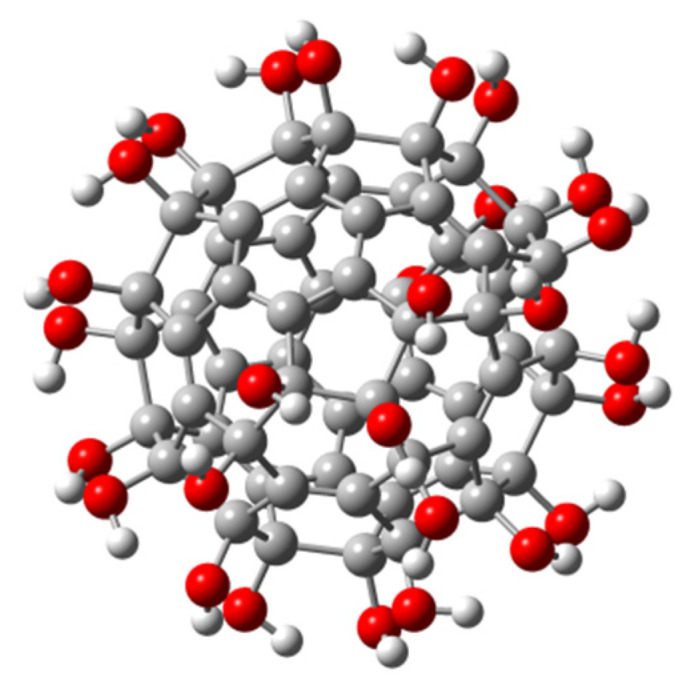
Structure diagram of fullerenol. Oxygen, carbon, and hydrogen atoms are marked in red, gray, and white, respectively.

**Figure 17 nanomaterials-12-02547-f017:**
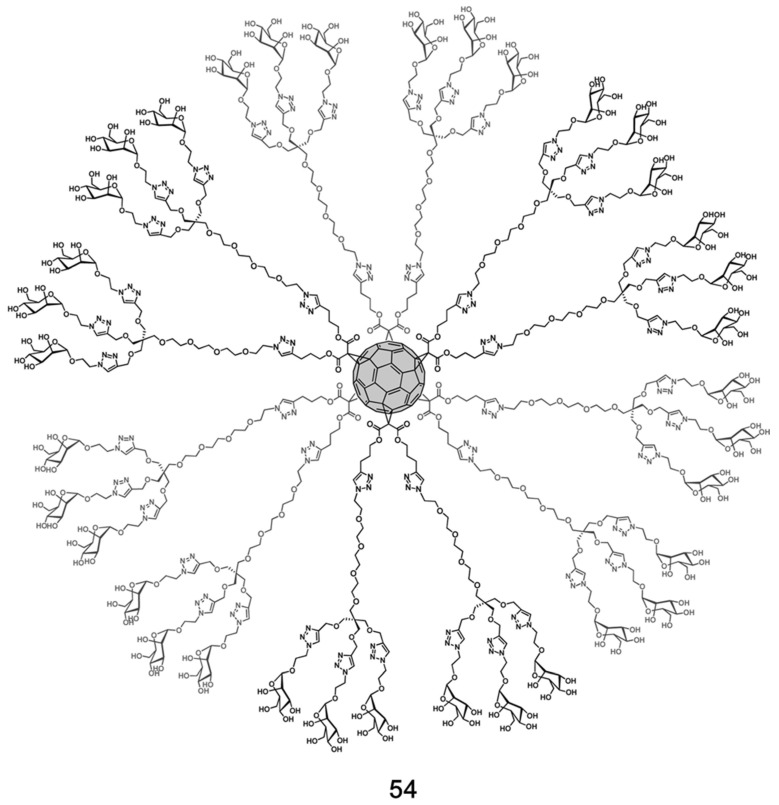
Structure diagram of glycofullerene (**54**). Adapted with permission from Ref. [78]. Copyright 2013 American Chemical Society.

**Figure 18 nanomaterials-12-02547-f018:**
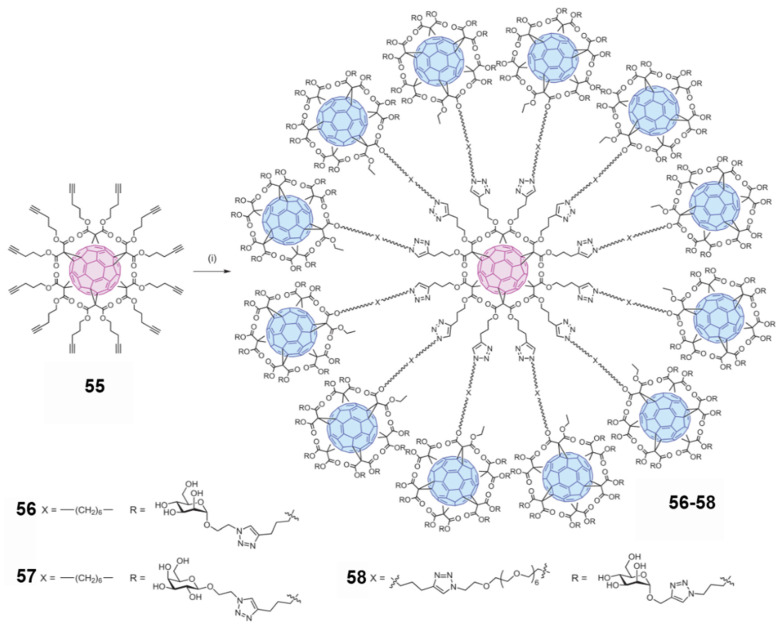
Synthesis of glycofullerenes **56****–58** [45]. Adapted with permission from Ref [45]. Copyright 2015 Springer Nature.

**Figure 19 nanomaterials-12-02547-f019:**
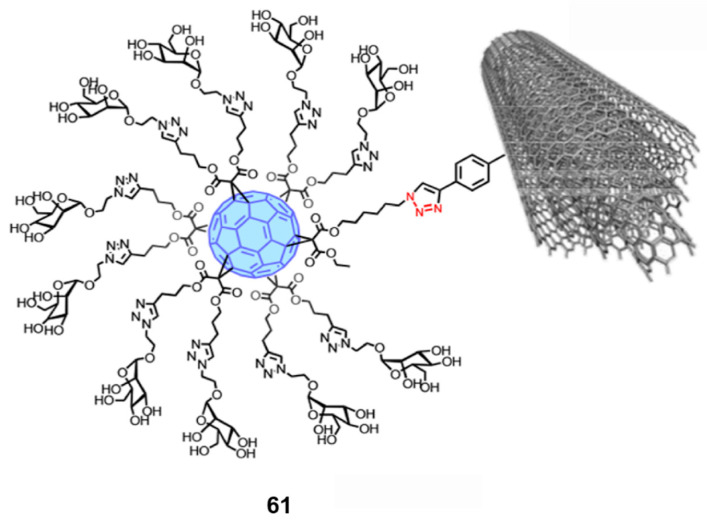
Structure diagram of nanoglycofullerene conjugate **61 [83]**. Adapted with permission from Ref. [83]. Copyright 2018 American Chemical Society.

**Figure 20 nanomaterials-12-02547-f020:**
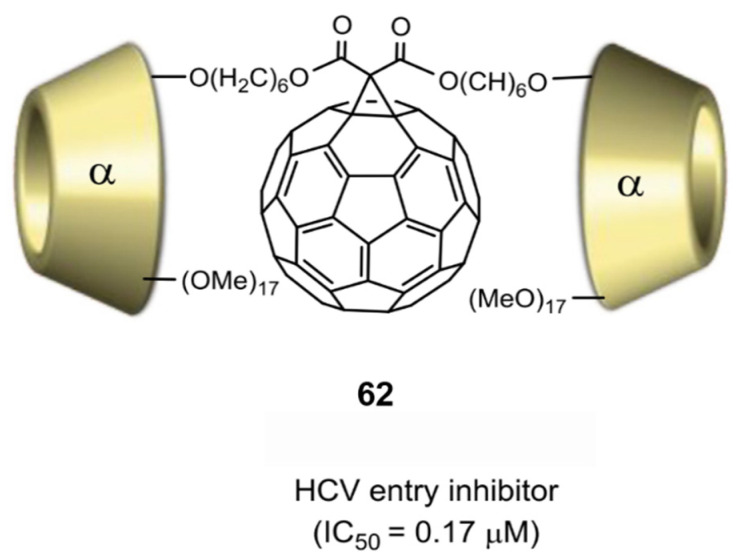
α-Cyclodextrin-C_60_ conjugate (**62**) [84]. Adapted with permission from Ref. [84]. Copyright 2012 Elsevier.

**Figure 21 nanomaterials-12-02547-f021:**
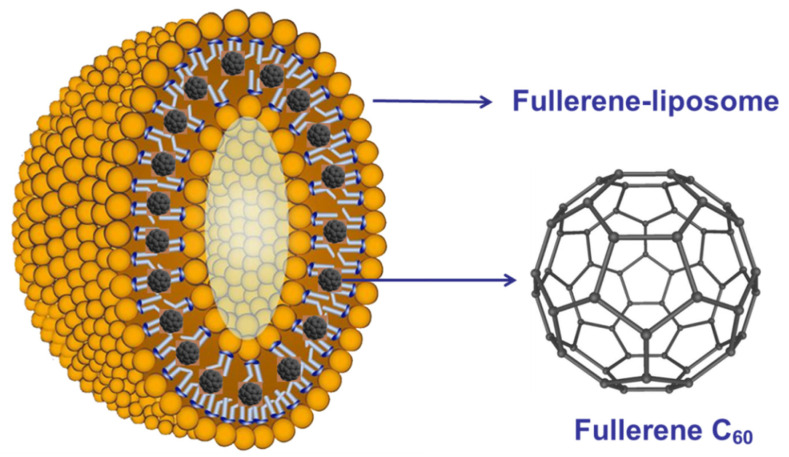
The structure of the fullerene liposome complex [87].

**Table 1 nanomaterials-12-02547-t001:** Anti-HIV activity and cytostatic toxicity of compounds (**11**–**23**) in CEM cell cultures.

Compound	EC_50_ (μM)		CC_50_ (μM)
	HIV-1	HIV-2	
**11**	>4	>4	44.3
**12** (*trans*-2)	>4	>4	7.2
**13** (*trans*-3)	>4	>4	7.63
**14** (*trans*-4)	>4	>4	7.4
**15** (*equatorial*)	>4	>4	9.6
**16** (*trans*-2)	0.40 ± 0.0	>4	4.79
**17** (*trans*-3)	0.96 ± 0.39	>4	3.02
**18** (*trans*-4)	2.60 ± 0.88	>4	13.2
**19** (*equatorial*)	1.60 ± 0.0	>4	6.59
**20** (*trans*-2)	>4	>4	-
**21** (*equatorial*)	>4	>4	-
**22** (*trans*-2)	2.01 ± 0.0	>4	-
**23** (*equatorial*)	>4	>4	-

**Table 2 nanomaterials-12-02547-t002:** Anti-HIV activity and cytostatic activity of compounds (**24**–**28**) in CEM cell cultures.

Compound	EC_50_ (μM)		CC_50_ (μM)
	HIV-1	HIV-2	
**24**	0.21 (±0.07)	0.2 to 1.0	2.93 (±1.20)
**25**	0.35 (±0.07)	0.70 (±0.42)	9.04 (±0.18)
**26**	1.08 (±0.57)	2.50 (±1.90)	12.5 (±7.54)
**27**	>25	>25	>125
**28**	2.50 (±0.71)	>10	28.7 (±1.27)

**Table 3 nanomaterials-12-02547-t003:** HIV-RT inhibitory activity and cytotoxicity of fullerene pyrrolidine derivatives (**29**–**41**).

Compound	R^1^	R^2^	R^3^	IC_50_ *	CC_50_ *
**29**	2-(*N*-Methylpyridinium)	–CH_2_COOC_2_H_5_	–C_2_H_5_	0.30	>50
**30**	2-(*N*-Methylpyridinium)	–H	–C_2_H_5_	0.33	>50
**31**	2-Pyridine	–CH_2_COOH	–H	0.20	>50
**32**	2-Pyridine	–H	–H	0.46	>50
**33**	3-(*N*-Methylpyridinium)	–CH_2_COOC_2_H_5_	–C_2_H_5_	0.25	>50
**34**	3-(*N*-Methylpyridinium)	–H	–C_2_H_5_	0.094	>50
**35**	3-Pyridine	–CH_2_COOH	–H	0.41	39.4
**36**	3-Pyridine	–H	–H	0.080	>50
**37**	4-(*N*-Methylpyridinium)	–CH_2_COOC_2_H_5_	–C_2_H_5_	0.74	>50
**38**	4-(*N*-Methylpyridinium)	–H	–C_2_H_5_	0.37	>50
**39**	4-Pyridine	–CH_2_COOH	–H	1.60	>50
**40**	4-Pyridine	–H	–H	0.80	>50
**41**	-	-	-	>500	>50
Nevirapine	-	-	-	3.52	-

* These values (~µM) are based on the average of three test results for each test compound.

**Table 4 nanomaterials-12-02547-t004:** IC_50_ and RIP values of different glycofullerene derivatives.

Compound	IC_50_/(nM)	Mannoses (No.)	RIP *
**56** (120 mannoses)	0.667	120	1.58 × 10^4^
**58** (120 mannoses)	20.375	120	5.2 × 10^2^
**54** (36 mannoses)	300	36	1.17 × 10^2^

* Relative inhibitory effect, calculated as (IC_50_)mono/IC_50_ * valency ((IC_50_)mono, IC_50_ of the monovalent compound; IC_50_ * valency, IC_50_ of the multivalent compound multiplied by the number of ligands present in the multivalent compound).

## Data Availability

Not applicable.

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
