# Peer review of "Progress in Antiviral Fullerene Research"

_nanomaterials, 2022, doi:10.3390/nano12152547_

Round 1

Reviewer 1 Report

The reviewer thinks this manuscript is a well organized review of the antiviral activity of fullerene derivatives and believes it will bring a lot of information to researchers who are interested in pharmaceutical applications of fullerenes. Some comments are:

1. The review classifies fullerene derivatives into six categories, but in several places this classification may not be appropriate. For example, even though amino acid and peptide derivatives are fine in the first classification, the reviewer does not feel comfortable including amines there. Especially for compound 10, it seems to be distinct from other compounds in this classification. It is also not clearly differentiated from the pyrrolidine type, which is in classification 2.

2. According to the literature survey conducted for the peer review, compound 34 is also mentioned in subsequent reports (Bioorg Med Chem Lett. 2021; 49:128267. doi: 10.1016/j.bmcl.2021.128267) as an inhibitor of HIV protease and HCV NS5B, so it should be mentioned in this review summarizing the antiviral effects of the fullerene derivatives. In addition, its exo-substituent without the fullerene core is also shown to have no inhibitory effect on the three enzymes. The reviewer would also like to add that 41 is not exo-substituent of 36, but exo-substituent of 34.

3. The resolution of some figures was very poor, with blurred or jagged structural formulas and text that could not be clearly identified (e.g., Fig. 22).

4. DODAB is not well known and should be written without abbreviation.

Author Response

Reviewer 1

The reviewer thinks this manuscript is a well organized review of the antiviral activity of fullerene derivatives and believes it will bring a lot of information to researchers who are interested in pharmaceutical applications of fullerenes. Some comments are:

Response: We thank the reviewer very much for his/her positive comments.

  1. The review classifies fullerene derivatives into six categories, but in several places this classification may not be appropriate. For example, even though amino acid and peptide derivatives are fine in the first classification, the reviewer does not feel comfortable including amines there. Especially for compound 10, it seems to be distinct from other compounds in this classification. It is also not clearly differentiated from the pyrrolidine type, which is in classification.

Response: We thank the reviewer for his/her kind suggestion. We fully agree with the reviewer. We have revised the first classification to amino acid, peptide, and primary amine derivatives. Compound 10, a piperazine derivative, has been classified as the second class with pyrrolidine because they all belong to cyclic amines.

  1. According to the literature survey conducted for the peer review, compound 34 is also mentioned in subsequent reports (Bioorg Med Chem Lett. 2021; 49:128267. doi: 10.1016/j.bmcl.2021.128267) as an inhibitor of HIV protease and HCV NS5B, so it should be mentioned in this review summarizing the antiviral effects of the fullerene derivatives. In addition, its exo-substituent without the fullerene core is also shown to have no inhibitory effect on the three enzymes. The reviewer would also like to add that 41 is not exo-substituent of 36, but exo-substituent of 34.

Response: We thank the reviewer for his/her professional comment and kind suggestion. We have supplemented the related description about compound 34 serving as an inhibitor of HIV protease and HCV NS5B and the ref. 59 (Bioorg Med Chem Lett, 2021, 49, 128267) in the revised manuscript. Additionally, compound 41 is indeed not the exo-substituent of compound 36, but the exo-substituent of compound 34, I have corrected the mistake.

  1. The resolution of some figures was very poor, with blurred or jagged structural formulas and text that could not be clearly identified (e.g., Fig. 22).

Response: We thank the reviewer for his/her kind suggestion. We have redrawn Figure 22 in the revised manuscript.

  1. DODAB is not well known and should be written without abbreviation.

Response: We thank the reviewer for his/her kind suggestion. We have revised the abbreviation of DODAB to dioctadecyldimethylammonium bromide.

Reviewer 2 Report

Comments from Reviewer

Title: Progress of fullerene antiviral research

The current form's presentation of methods and scientific results is satisfactory for publication in the Nanomaterials journal. The minor and significant drawbacks to be addressed can be specified as follows:
1.    Line 50, "39,43". 39-43?
2.    (i) Line 94. Wudl et al. ---> Friedman et al. (ii) Line 100. Wudl et al. ---> Sijbesma et al. Check the manuscript. See, for example – Prato et al. The first author should be given!!!
3.    Lines 114 and 115. 2 and 3 ---> 2 and 3 (Fig. 3). The form of the notation proposed by the authors is not entirely transparent.
4.    Fig. 3 and others. Copyright is another problem. Authors must have publishers' permission to use the figures. Fig. 3 is taken from [52] – see Fig. 2 from https://pubs.acs.org/doi/10.1021/ol000217y In addition to the information about the permission to use the respective figures, please also include the reference (i.e. [52]) in the figure captions.
5.    Literature should also be standardized: the size of letters in the titles of journals, initials of names, and the size of letters in the titles of articles.

Sincerely,
     The reviewer.

Author Response

Reviewer 2:

The current form's presentation of methods and scientific results is satisfactory for publication in the Nanomaterials journal. The minor and significant drawbacks to be addressed can be specified as follows:

Response: We thank the reviewer very much for his/her positive comments. In order to further improve the quality of the manuscript, we have asked the language editor to polish it.

  1. Line 50, "39,43". 39-43?

Response: We thank the reviewer for his/her careful revision. Ref. 39-40 are correct, we have revised the mistake.

  1. (i) Line 94. Wudl et al. ---> Friedman et al. (ii) Line 100. Wudl et al. ---> Sijbesma et al. Check the manuscript. See, for example – Prato et al. The first author should be given!!!

Response: We thank the reviewer for his/her kind suggestion. We have checked all the manuscript and revised some of them to the first author or revised the “et al.” to “and coworkers”.

  1. Lines 114 and 115. 2 and 3 ---> 2 and 3 (Fig. 3). The form of the notation proposed by the authors is not entirely transparent.

Response: We thank the reviewer for his/her kind suggestion. We have added the notation of figure 3.

  1. Fig. 3 and others. Copyright is another problem. Authors must have publishers' permission to use the figures. Fig. 3 is taken from [52] – see Fig. 2 from https://pubs.acs.org/doi/10.1021/ol000217y In addition to the information about the permission to use the respective figures, please also include the reference (i.e. [52]) in the figure captions.

Response: We thank the reviewer for his/her kind suggestion. For figure 3-4 and figure 17-20, we have obtained the permission from publishers, and included references.

  1. Literature should also be standardized: the size of letters in the titles of journals, initials of names, and the size of letters in the titles of articles.

Response: We thank the reviewer for his/her kind suggestion. We have checked all references and revised the format of all references according to the requirements of Nanomaterials.

Round 2

Reviewer 2 Report

Congratulations on a great job. The author has made a substantial improvement to this article. The manuscript can be accepted for publishment in the present form.